# Flood Monitoring in the Middle and Lower Basin of the Yangtze River Using Google Earth Engine and Machine Learning Methods

**Jingming Wang** [1], **Futao Wang** [2,3,*], **Shixin Wang** [2,3], **Yi Zhou** [2,3], **Jianwan Ji** [4], **Zhenqing Wang** [2,3], **Qing Zhao** [2] **and Longfei Liu** [5]

1. Jiangsu Provincial Key Laboratory of Geographic Information Science and Technology, Key Laboratory for Land Satellite Remote Sensing Applications of Ministry of Natural Resources, School of Geography and Ocean Science, Nanjing University, Nanjing 210023, China
2. Aerospace Information Research Institute, Chinese Academy of Sciences, Beijing 100094, China
3. University of Chinese Academy of Sciences, Beijing 100049, China
4. School of Geography Science and Geomatics Engineering, Suzhou University of Science and Technology, Suzhou 215009, China
5. National Disaster Reduction Center of China, Beijing 100124, China
* Correspondence: wangft@aircas.ac.cn; Tel.: +86-134-264-025-82

**Abstract:** Under the background of intensified human activities and global climate warming, the frequency and intensity of flood disasters have increased, causing many casualties and economic losses every year. Given the difficulty of mountain shadow removal from large-scale watershed flood monitoring based on Sentinel-1 SAR images and the Google Earth Engine (GEE) cloud platform, this paper first adopted the Support Vector Machine (SVM) to extract the water body information during flooding. Then, a function model was proposed based on the mountain shadow samples to remove the mountain shadows from the flood maps. Finally, this paper analyzed the flood disasters in the middle and lower basin of the Yangtze River (MLB) in 2020. The main results showed that: (1) compared with the other two methods, the SVM model had the highest accuracy. The accuracy and kappa coefficients of the trained SVM model in the testing dataset were 97.77% and 0.9521, respectively. (2) The function model proposed based on the samples achieved the best effect compared with other shadow removal methods with a shadow recognition rate of 75.46%, and it alleviated the interference of mountain shadows for flood monitoring in a large basin. (3) The flood inundated area was 8526 km$^2$, among which, cropland was severely affected (6160 km$^2$). This study could provide effective suggestions for relevant stakeholders in policy making.

**Keywords:** flood monitoring; mountain shadows; Google Earth Engine; support vector machine; middle and lower basin of the Yangtze River





## 1. Introduction

Floods are one of the most serious natural disasters in the world, causing huge casualties and economic losses worldwide every year [1,2]. In recent years, with global climate warming, the frequency and intensity of floods have become higher. According to the "China Flood and Drought Disaster Prevention Bulletin" released by the Ministry of Water Resources (http://www.mwr.gov.cn (accessed on 12 April 2022)), floods in 2020 caused 230 deaths across the country and direct economic losses of CNY 266.98 billion. Therefore, timely and accurate monitoring of floods and analyzing the evolution trends of floods are of great significance to disaster emergency management in China [3].

At present, remote sensing technology has gradually become the main means for flood monitoring due to its advantages of wide coverage and low revisit period [4,5]. Depending on the detection method, it can be divided into optical remote sensing monitoring and radar remote sensing monitoring. However, since floods are usually accompanied by cloudy or

rainy weather, the earth observation of optical remote sensing satellites are hindered, so it is difficult to obtain clear and cloud-free optical images [6]. On the contrary, Synthetic Aperture Radar (SAR) plays an increasingly important role in flood monitoring because of its all-day and all-weather working ability, and it is not easily affected by cloudy and rainy weather [7,8].

To date, flood monitoring methods based on SAR data mainly include the thresholding-based method [9,10], object-oriented method [11], active contour method [12], and machine learning method [13,14]. Among them, the thresholding-based method is the most used in flood monitoring. Although its running is fast and the principle is simple [15], it is difficult to meet the accuracy requirements in the face of uneven image grayscale and large flood area range [16]. The object-oriented method can utilize features such as texture and shape of the image, and although good results are achieved, the scale parameters for segmentation and classification depend on experience [17]. The active contour method can make full use of the color features and edge information of the image, but the speckle noise in the image and the complex calculations hinder its application for flood monitoring in large basins [18]. In recent years, the machine learning method has been gradually applied to flood monitoring based on SAR images, and has achieved high extraction accuracy [19]. It can make full use of the feature information of images, and the trained model can be used for multiple images of the same type, which is suitable for batch processing. Considering the large geographic range of the study area, the noise impact of SAR images, and the complex and diverse flood scenes, we used the machine learning method to monitor large-scale floods.

As the available remote sensing data become more abundant and the amount of data become larger, offline processing takes a lot of time, so it is difficult to meet the demands of disaster emergency monitoring. The emergence of cloud platforms such as Google Earth Engine (GEE) has solved the problem of long processing times and large amounts of calculations for remote sensing images [20]. GEE is a cloud computing platform specialized in processing remote sensing images. It stores the main open-access remote sensing image datasets from the past 40 years, such as Landsat, Sentinel, Modis series data, etc. [21]. With its powerful computing capabilities, massive free remote sensing data, and many built-in algorithms, GEE provides important and technical support for flood monitoring in large basins [22]. DeVries et al. [23] used all available Sentinel-1 images, combined with historical Landsat and other auxiliary data, to quickly extract inundation information during floods based on the GEE platform. Qiu et al. [24] used the 66 Sentinel-1 images of the GEE platform to study the floods in the Pearl River Basin from 2017 to 2020 using the Otsu thresholding method. Jia et al. [25] obtained the spatial and temporal distribution pattern of floods in the Chaohu Lake Basin from 2015 to 2020 based on Sentinel-1 images in the GEE platform, and analyzed the impact of floods on cropland and buildings. However, most of the existing studies focus on flooding in small basins, and flood monitoring in large-scale basins is affected by mountain shadows.

In addition, due to the uneven grayscale distribution in SAR images [18], easy confusion of water body and mountain shadows [26], and limited computing resources [24], the existing large-scale watershed flood monitoring is not very accurate. Therefore, based on the GEE cloud computing platform, in this work, we used all Sentinel-1 images during the flood disasters period in the study area, first used the SVM method to extract the flood water body information. Then, we proposed a novel function model to remove the mountain shadows from the flood maps. Finally, we post-processed the results to analyze the flood disasters in the MLB in 2020.

## 2. Materials and Methods

### 2.1. Study Area

The MLB is located in the south of the Qinling Mountain and Huaihe River, with the geographic range of 24°29′~34°11′ N and 106°5′~121°54′ E (Figure 1). The MLB covers an area of about 800,000 km$^2$, spanning Hubei, Hunan, Jiangxi, Anhui, Jiangsu, Zhejiang,

Shanghai, and other provinces and cities. The MLB is an important industrial base and one of the most economically developed regions in China. In addition, the MLB is an important production base for grain, edible oil, and cotton, known as the "land of fish and rice", with many lakes such as Poyang Lake, Dongting Lake, Taihu Lake, and Chaohu Lake [27]. The MLB is high in the west and low in the east, with mountains in the west that hinder flood monitoring. From June to July 2020, there were 10 consecutive heavy rainfalls in the MLB and severe floods occurred in the basin [28], which inundated large areas of cropland and infrastructure, and causing serious losses to people's lives and property safety.

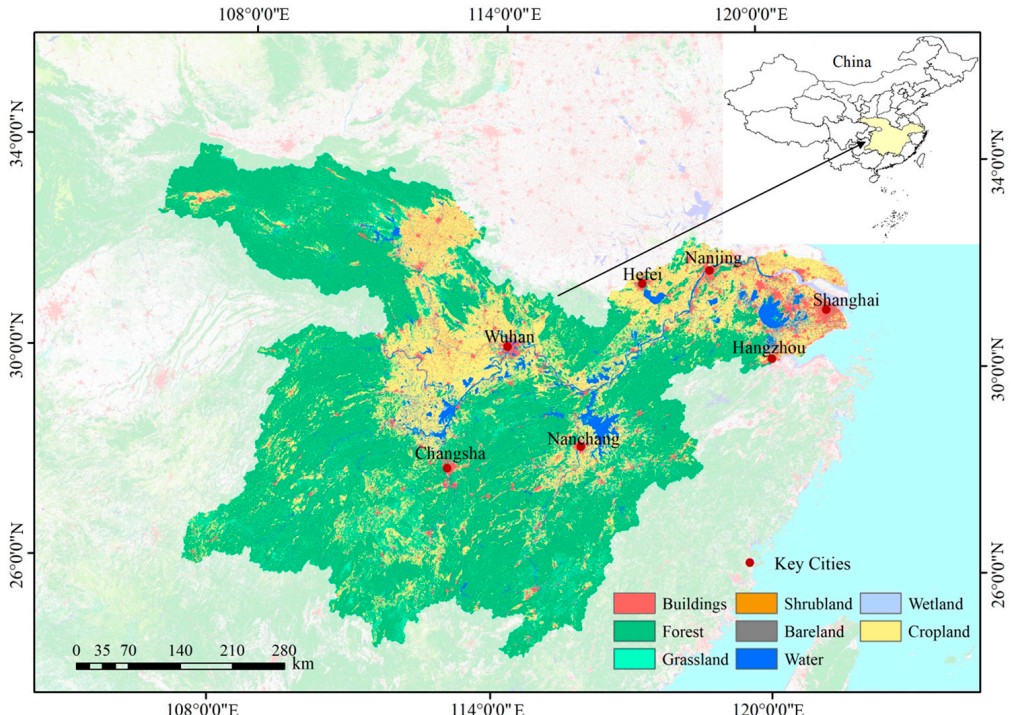

**Figure 1.** Location of the MLB in China.

## 2.2. Data Source and Pre-Processing

### 2.2.1. Sentinel-1 SAR Data

The Sentinel-1 SAR satellites, developed by the European Space Agency (ESA), are in a sun-synchronous orbit and consist of two satellites, A and B. The lowest revisit period of the two satellites is 6 days, the maximum width is 400 km, and the highest spatial resolution reaches 5 m. There are four imaging modes with all-weather earth observation capability [29]. With the advantages of good data quality and high resolution, Sentinel-1 SAR images have gradually become an important data source for flood monitoring.

We used 320 Sentinel-1 images stored in the GEE cloud service with an imaging time from 16 June 2020 to 8 September 2020 to cover the entire flood stage. Five floods occurred in the Yangtze River basin in 2020, on 2 July, 12 July, 26 July, 14 August, and 17 August. We chose images after 16 June that could cover the study area before flooding, and images before 8 September to cover the study area after flooding. The image coverage is shown in Figure 2. The preprocessing operations of Sentinel-1 data on the GEE platform include thermal noise removal, radiometric calibration, and terrain correction [30]. Additionally, we performed mean filtering on the Sentinel-1 data, using the filter window size 3 × 3 for suppressing speckle noise in the image. After preprocessing, we obtained the backscattering maps with a resolution of 10 m in the study area.

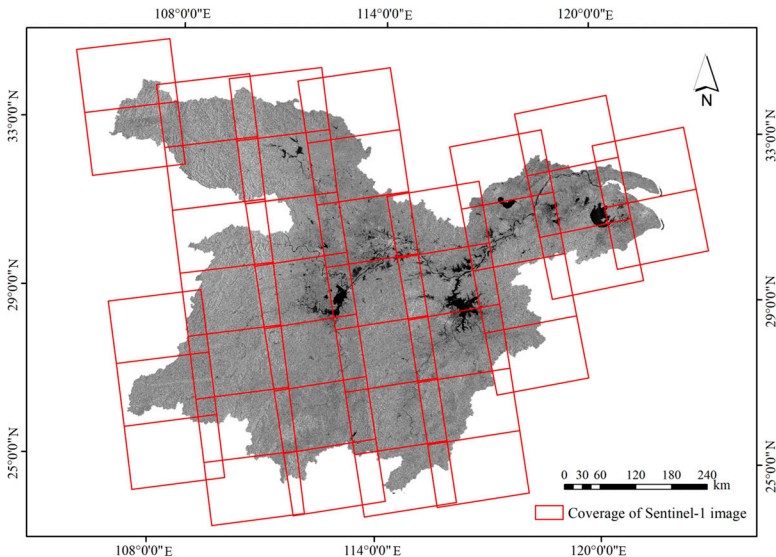

**Figure 2.** The geographic coverage of Sentinel-1 images in the study area.

### 2.2.2. Sentinel-2 Optical Data

The Sentinel-2 high-resolution multispectral imaging satellites developed by the ESA, consist of two satellites A and B. The revisit period of a single satellite is 10 days, with 13 spectral bands, and its spatial resolution is up to 10 m. In this paper, we used the Sentinel-2 optical images to assist the interpretation of Sentinel-1 images, so as to accurately generate samples for machine learning. The specific information is shown in Table 1.

**Table 1.** Detailed information of Sentinel-2 images.

| Index | Image Name | Date | Cloud Cover |
|:---:|:---:|:---:|:---:|
| 1 | S2_SR/20200816T023549_20200816T024732_T51RTQ | 16 August 2020 | 1.31% |
| 2 | S2_SR/20200819T024549_20200819T025732_T50RPV | 19 August 2020 | 1.16% |
| 3 | S2_SR/20200828T031539_20200828T032736_T49RDL | 28 August 2020 | 1.28% |
| 4 | S2_SR/20200828T031539_20200828T032736_T49RDM | 28 August 2020 | 2.41% |
| 5 | S2_SR/20200830T030551_20200830T031738_T49RFN | 30 August 2020 | 2.58% |
| 6 | S2_SR/20200901T025549_20200901T030025_T50RKU | 1 September 2020 | 2.21% |
| 7 | S2_SR/20200904T030549_20200904T031747_T49SES | 4 September 2020 | 0.72% |
| 8 | S2_SR/20200906T025551_20200906T030731_T50RMT | 6 September 2020 | 1.68% |

### 2.2.3. Land Use Data

The land use data reflects the human intervention in land resources. We obtained the land use data from the National Fundamental Geographic Information Center (http://www.globeland30.org (accessed on 20 June 2022)) with a spatial resolution of 30 m. The eight categories of land use in the study area are buildings, shrubland, wetland, forest, bare land, cropland, grassland, and water, as shown in Figure 1. By analyzing the specific categories of inundated land use, the research analyzed the damage situation of the flood disasters.

### 2.2.4. DEM Data

The Digital Elevation Model (DEM) represents ground elevation information in a discrete raster. The DEM used in this paper was provided by Geospatial Data Cloud site, Computer Network Information Center, Chinese Academy of Sciences (http://www.gscloud.cn (accessed on 25 July 2022)) with a spatial resolution of 30 m. We used DEM data to calculate slope data, and proposed a linear function model based on DEM data, slope data, and shadow samples to remove mountain shadows that were easily confused with bodies of water in SAR images.

### 2.3. Methods

Machine learning algorithms commonly used for flood monitoring are SVM, Random Forest (RF), Decision Tree, Convolutional Neural Network (CNN), etc. For SVM, RF, and decision tree algorithms, the input to the model is point data, and the predicted result is also point data. For CNN, the input to the model is the small size feature maps, and the predicted result is also the feature maps with the original size [14]. The principle of machine learning models, including deep learning models, in extracting flood maps is to let the model learn given the sample dataset, so that the trained model can classify images by pixels [31].

SVM was proposed by Vapnik in 1963. It is one of the most popular algorithms in the field of machine learning and is widely used in remote sensing classification tasks [32]. The goal of SVM is to find an optimal boundary for classification in the feature space composed of sample points, which is called a hyperplane, so that the sample points of each category can be optimally separated. The common kernel function types of SVM are Linear, Poly, RBF, Sigmoid, etc. When the input data can linearly separate, the kernel function of SVM is usually Linear. On the contrary, researchers usually consider upgrading the dimensionality of the input data or using Poly, RBF, or Sigmoid as the kernel function. The SVM algorithm has the advantages of small sample learning, anti-noise, high learning efficiency, and good generalization, and has achieved good results in the field of remote sensing supervised classification [33].

In this paper, the sample points with the attributes of VH features, VV features, and derived SDWI features [34] of Sentinel-1 images were input to the SVM model for remote sensing classification. After the SAR data preprocessing was completed, the Sentinel-1 dual-polarization features and the derived SDWI features were first combined into the new three-band images. Then, the sample data was divided into training, validation, and testing datasets, which was input to the SVM model for training and to evaluate the accuracy of the model. Furthermore, all three-band images in the study area were predicted based on the trained SVM model, and the predicted results were post-processed. Finally, flood disaster analysis was performed based on all water body information distribution maps. The specific flowchart of this paper is shown in Figure 3.

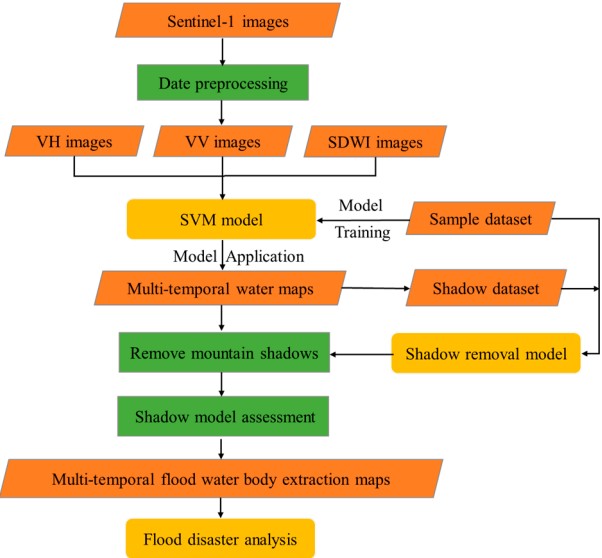

**Figure 3.** The flowchart of this paper.

### 2.4. Sample Generation

Whether the trained SVM model can make accurate predictions depends on the construction of the sample dataset, and the randomly distributed speckle noise in the SAR images [35] hinders the production of samples. In order to ensure the validity and

accuracy of the sample data, the training samples and testing samples were visually interpreted from Sentinel-1 images with the aid of Sentinel-2 optical remote sensing images, where the imaging time of Sentinel-2 and Sentinel-1 images was controlled within one day, and the Sentinel-2 images were below 3% cloud cover. The study produced a total of 50,000 training samples and 20,000 testing samples, which included 14,423 water samples and 35,577 non-water samples in the training samples, and 8842 water samples and 11158 non-water samples in the testing samples. The training samples came from six geographic regions and different time images (Figure 4), thus guaranteeing the robustness and generalization of the SVM model.

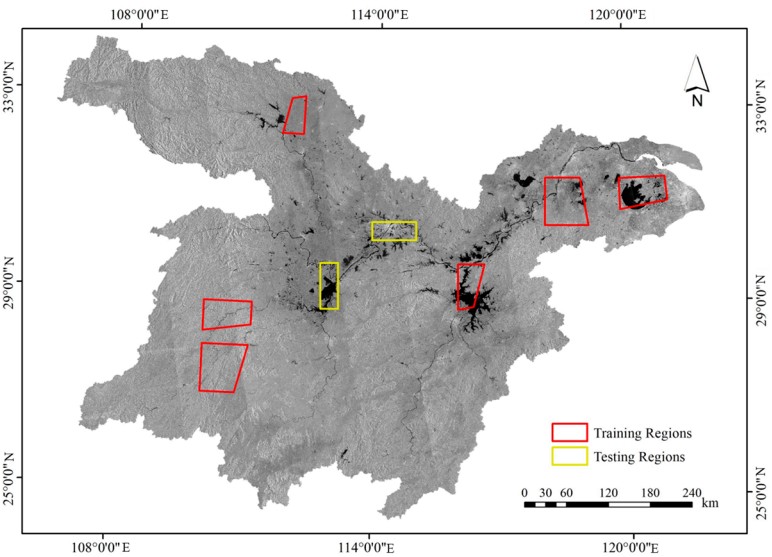

**Figure 4.** Training regions and testing regions in the study area.

Since the training samples were difficult to classify on the two-dimensional space composed of VH and VV polarization modes (Figure 5) and considering the limited polarization data of Sentinel-1, we introduced SDWI features to enrich the data features [36]. Firstly, the SDWI features were constructed using preprocessed Sentinel-1 dual polarization data. Then, VH features, VV features, and SDWI features were combined into the three-band images. Finally, the training samples with VH–VV–SDWI feature information were input into the SVM model for training. SDWI was the SAR image water index feature proposed by Jia in 2018 [34], the formula is as follows:

$$K_{SDWI} = ln \, (10 \times VH \times VV) \tag{1}$$

where $K_{SDWI}$ refers to the result value of band math operation, and $VH$ and $VV$ refer to Sentinel-1 dual polarization data. The SDWI reference drew on the vegetation index NDWI, which used the band math operation between Sentinel-1 dual polarization data to enhance the water body information, and achieved a good water body information extraction effect.

### 2.5. Accuracy Assessment

In this paper, we chose the accuracy and kappa coefficient to quantitatively evaluate the performance of the trained SVM model. The accuracy represents the ratio of the number of correctly classified pixels to the total number of pixels. It is simple to calculate and is usually expressed as a percentage. The higher the value, the better the performance of the model. However, due to the uneven number of samples of each category in the classification process, the high accuracy cannot indicate the degree of classification accuracy of each category. Meanwhile, the kappa coefficient is a comprehensive evaluation index to measure the performance of the model, which is a better reference, and its range is between −1 and 1. The higher the value, the better the model prediction.

$$acc = (TP + TN)/(TP + FN + FP + TN) \tag{2}$$

$$pe = \sum_{i=0}^{k}(a_i b_i)/N^2 \tag{3}$$

$$Kappa = (acc - pe)/(1 - pe) \tag{4}$$

where *acc* and *Kappa* refer to the accuracy and kappa, respectively. *TP* refers to the number of real water body pixels predicted by the model as water body. *FP* refers to the number of real non-water body pixels predicted by the model as water body. *FN* refers to the number of real water body pixels predicted by the model as non-water body. *TN* refers to the number of real non-water body pixels predicted by the model as non-water body. *k* refers to the number of sample classification categories. *N* refers to the total number of samples, $a_i$ refers to the number of real samples in each category, and $b_i$ refers to the number of samples in each category predicted by the SVM model.

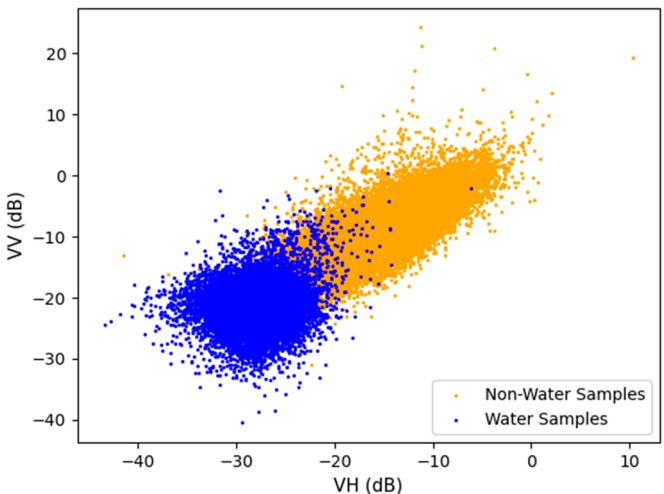

**Figure 5.** The distribution of training samples in VH–VV two-dimensional space.

## 3. Experiment and Results

### 3.1. Model Training

The study randomly divided 50,000 samples into training samples and validation samples according to the ratio of 80% and 20% on the GEE cloud platform, and then input 40,000 training samples and 10,000 validation samples into SVM model for training. Through multiple experiments, this study finally obtained the appropriate training parameters, the SVM type was set to C_SVC, and Linear was selected as the kernel function type. The SVM model finally converged after iterative training and achieved 98.14% and 98.03% on the training samples and the validation samples, respectively, and its corresponding kappa coefficients were 0.9548 and 0.9547, respectively.

### 3.2. Model Testing

Given that samples from different geographic regions could better measure the flood water extraction ability of the SVM model, 20,000 testing samples came from two geographic regions. The accuracy of the SVM model on the testing samples was 97.77%, and the kappa coefficient was 0.9521 (Table 2). The results showed that the trained SVM model had strong robustness and generalization.

### 3.3. Postprocessing

The 320 three-band images in the study area were input into the trained SVM model for prediction of a spatial distribution map of water body information during the floods in the MLB. Since the backscattering coefficients of water bodies and mountain shadows

on SAR images are similar, they are easily confused, resulting in low accuracy of remote sensing intelligent extraction [26,37]. Therefore, we proposed a function model with the help of DEM and derived slope data in the study area to remove the influence of mountain shadows. First, with the aid of Sentinel-1 images, 41,214 shadow points were visually interpreted from the water extraction distribution map. The 41,214 shadow points were identified as water bodies by the SVM model, but were actually non-water bodies. After that, the attribute values of the corresponding locations of the shadow points were extracted from the DEM and slope data. Subsequently, the shadow points and the water samples in the training dataset were plotted in the two-dimensional space of elevation and slope, as shown in Figure 6. It could be seen that the water body and shadow samples were separable in the two-dimensional space of elevation and slope. Combined with previous research on removing mountain shadows [38,39], this study proposed four methods for removing shadows, as shown in Table 3 and Figure 7.

**Table 2.** Accuracy results of different sample datasets.

| Data | Accuracy (%) | Kappa |
|---|---|---|
| Training samples | 98.14 | 0.9548 |
| Validation samples | 98.03 | 0.9547 |
| Testing samples | 97.77 | 0.9521 |

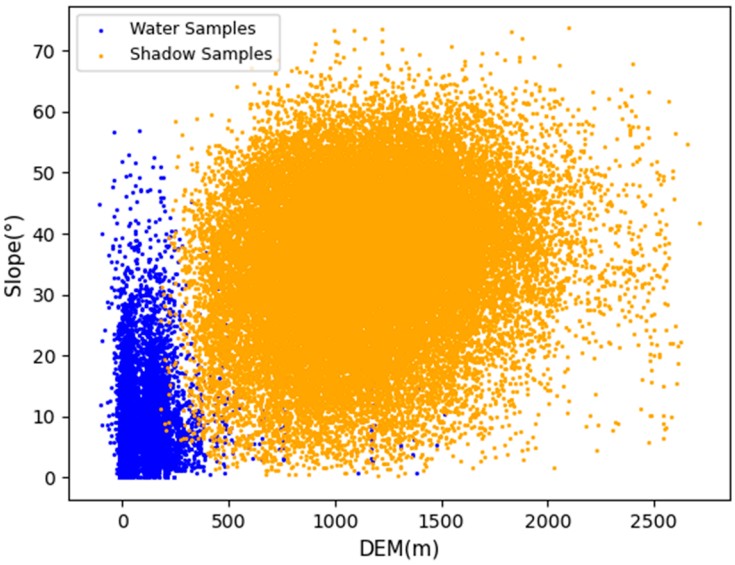

**Figure 6.** The water body and shadow samples in the two-dimensional space of elevation and slope. The number of water body and shadow samples were 23,265 and 41,214.

**Table 3.** The schematic table of mountain shadow removal methods.

| Method | Classification | Shadow Recognition Rate * |
|---|---|---|
| 1 | The water body elevation and slope thresholds were 1518 and 56.978 | 0.706% |
| 2 | The water body elevation and slope thresholds were 318 and 29.2278 | 70.54% |
| 3 | The water body elevation and slope thresholds were 414 and 34.0374 | 55.94% |
| 4 | $Y = -0.0324x + 59.5059$ | 75.46% |

* The shadow recognition rate refers to effectiveness of the method in recognizing shadows, and higher value indicates that the method is better.

Methods 1 to 3 were threshold methods. The threshold value in method 1 was determined according to the maximum elevation and maximum slope of the water samples. If the elevation and slope at any location in the study area were both higher than the threshold value, this location was identified as mountain shadow. The maximum threshold

was applied to the shadow samples, which could remove 0.706% of the shadow samples (Figure 7a). The threshold value in method 2 was determined according to the proportion of the elevation and slope values of the water samples arranged from low to high to 99%, and the shadow recognition rate reached 70.54% (Figure 7b). Similarly, the threshold value in method 3 was determined according to 99.5%, and the shadow recognition rate was 55.94% (Figure 7c). Method 4 was the function method, which was proposed according to the water sample points with the maximum elevation (DEM: 1518 m, slope: 10.3082°) and the maximum slope (DEM: 78 m, slope: 56.978°), and its shadow recognition rate was 75.46% (Figure 7d). The function model could alleviate the interference of mountain shadows without reducing the accuracy of flood monitoring. Therefore, this paper selected the linear function model proposed by method 4 to remove the mountain shadows. The formula was as follows:

$$Y = -0.0324x + 59.5059 \tag{5}$$

where $x$ refers to the elevation at any location in the study area. If the slope value at the corresponding location was higher than the $Y$ value, the function model identified this location as mountain shadow, otherwise it was not mountain shadow. For each water body pixel identified by the SVM model, we obtained the elevation value and slope value at the pixel, and input them into the shadow removal model, which then determined whether the pixel was water body or mountain shadow.

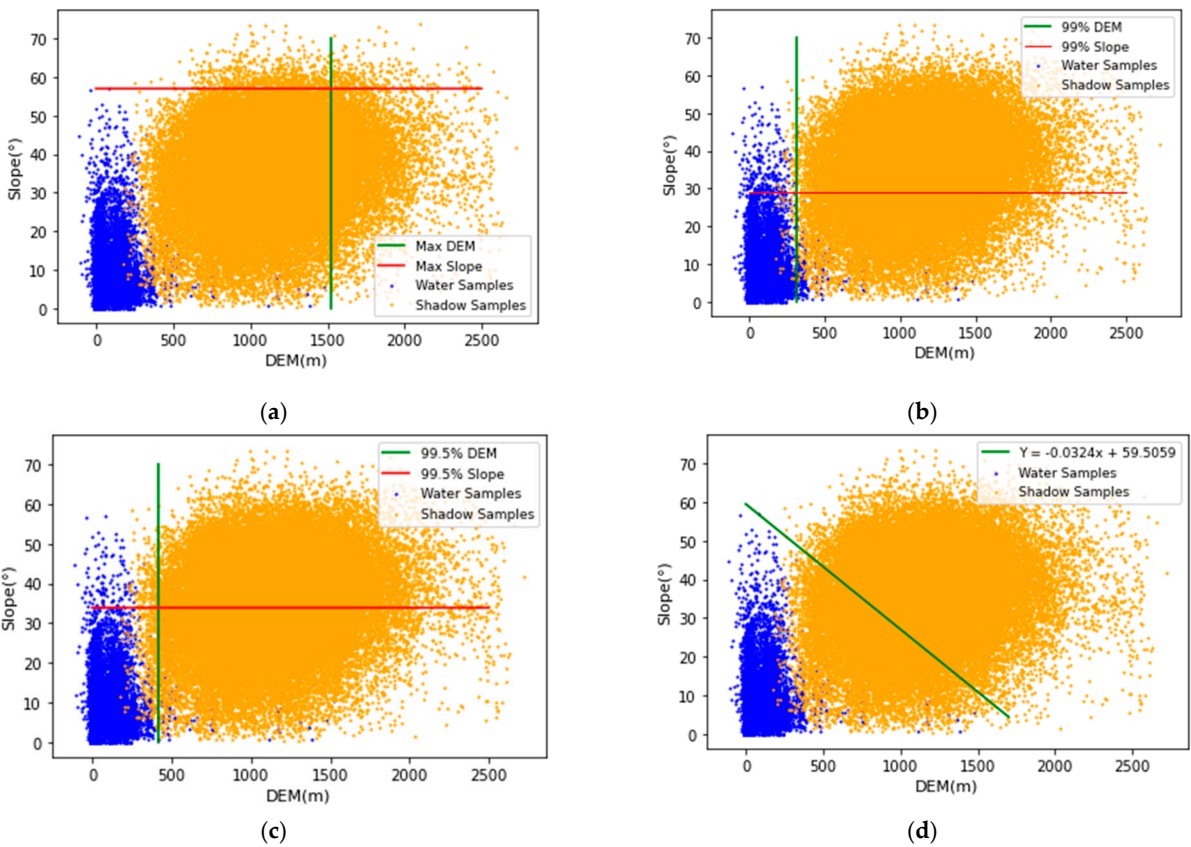

**Figure 7.** The schematic figures of mountain shadow removal methods. (**a**) Method 1, (**b**) Method 2, (**c**) Method 3, (**d**) Method 4.

The study applied the linear function model to all the water body information distribution maps to obtain the water body distribution maps after removing the mountain shadows. Due to the influence of factors such as image noise, wind, waves, bridges, and ships, the extracted water body contained some holes or voids. Therefore, this paper used mathematical morphological operations to post-process the results after removing the mountain shadows. Mathematical morphological operations mainly included remov-

ing small objects and small holes. We used the functions provided by the scikit-image package, set the minimum size of removing small objects operation to 100 and the area threshold of removing small holes operation to 100, and chose the 4-neighborhood mode to post-process the results. The post-processing results were shown in Figure 8. After post-processing the results predicted by the trained SVM model, the boundaries of rivers and lakes were completely extracted, while the interference of mountain shadows was effectively alleviated.

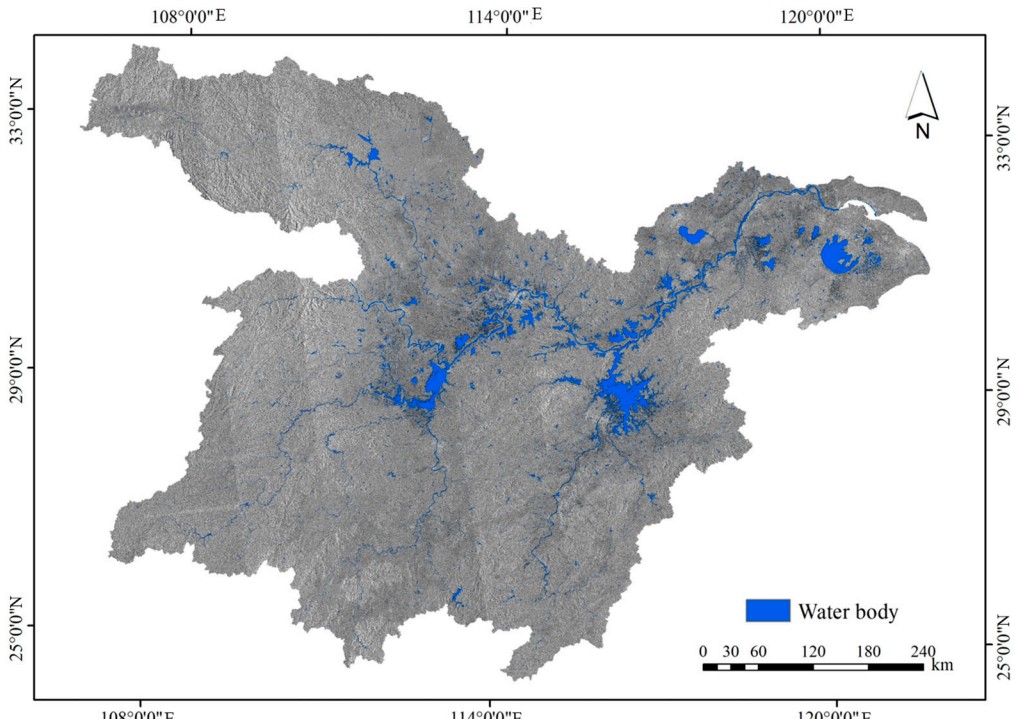

**Figure 8.** Detailed map after post-processing showing the mosaic results of water body from 10 July 2020 to 20 July 2020.

## 4. Discussion

### 4.1. Comparison of Different Methods

In order to validate the accuracy of the SVM model, we compared its results with the extraction results of the RF method and Otsu method. The RF method is also one of the most popular algorithms in the field of machine learning. We set the tree number of RF to 100. The Otsu method is the most used thresholding-based method. We computed the Otsu threshold for the VH polarization mode of the Sentinel-1 images. The comparison results of the three methods are shown in Table 4. The results showed that the SVM model had the highest accuracy and the accuracy of the RF model was second only to the SVM model. The Otsu method had the lowest accuracy. The accuracy of the thresholding-based method was lower than that of the machine learning method. The effectiveness of this paper in extracting flood water body using the SVM model was proven.

**Table 4.** Accuracy comparison of different methods using the testing samples.

| Method | Accuracy (%) | Kappa |
|--------|--------------|-------|
| SVM | 97.77 | 0.9521 |
| RF | 96.79 | 0.9346 |
| Otsu | 91.92 | 0.8449 |

*4.2. Analysis of Shadow Removal Model*

This paper proposed a function model to remove mountain shadows from the water body extraction maps, but how much did this model improve the flood monitoring capability? Therefore, we analyzed the function model from qualitative and quantitative perspectives.

4.2.1. Qualitative Analysis

In order to qualitatively evaluate the ability of the function model to remove the mountain shadows, this paper selected four geographic areas to display the details after removing the mountain shadows, as shown in Figure 9. The left column of Figure 9 is the Landsat 8 optical remote sensing image of the four geographic regions. The middle column contains the corresponding Sentinel-1 SAR images. The right column is the result after removing the mountain shadows, in which the blue elements represented the water body after removing the mountain shadows, and the green elements represented shadows recognized by the function model. It could be seen that the shadows formed by tall mountains obscuring the radar beam were identified (region 1 and region 3), but the shadows caused by low mountains were partially identified, and some shadows were not extracted (region 2 and region 4). The results showed that the linear function model proposed in this paper had a good effect in removing the mountain shadows, and it could mitigate the interference of mountain shadows in large-scale watershed flood monitoring.

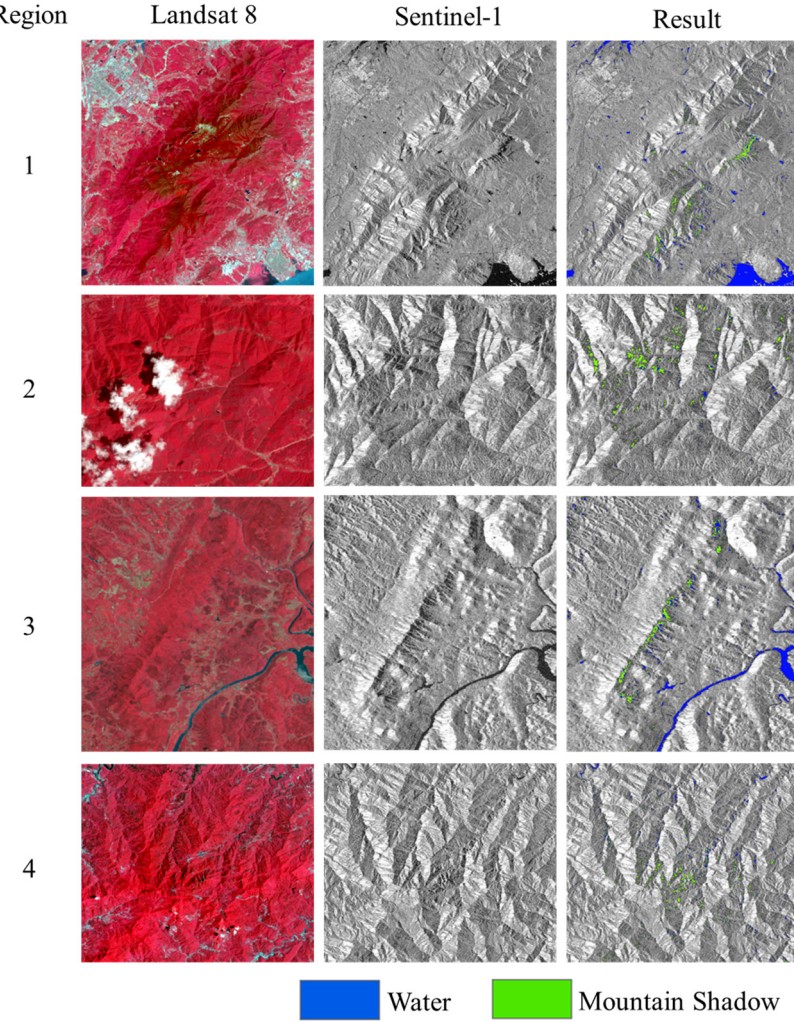

**Figure 9.** Detailed maps after removing the mountain shadows.

### 4.2.2. Quantitative Analysis

We selected a geographic region with a pixel size of 1319 × 2058 to quantitatively evaluate the effect of the shadow removal model. The selected region is in the west of the study area, and we randomly generated 5000 points to quantify the shadow model in the selected area. The water body results before and after removing mountain shadows were compared with actual water (Figure 10), and the comparison results are shown in Table 5. The accuracy and kappa coefficients in the selected region before removing mountain shadows was 93.06% and 0.9173, respectively. After removing mountain shadows, they were 95% and 0.9315, respectively. The accuracy and kappa coefficients after removing mountain shadows were improved by 1.94% and 0.0142, respectively. It could be seen that the function model was helpful for improving the flood monitoring ability.

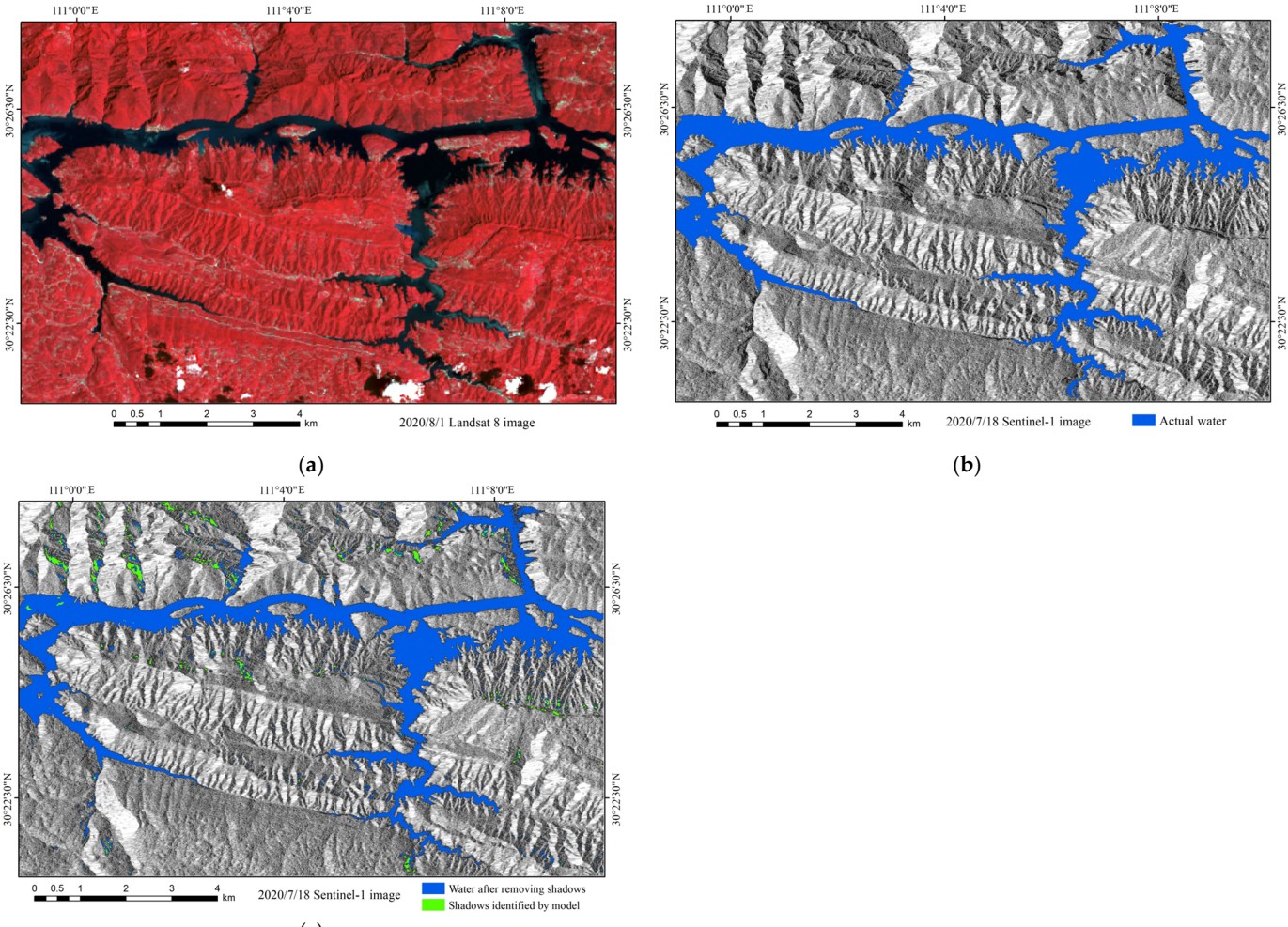

**Figure 10.** The results of removing mountain shadows in the selected region. (**a**) Landsat 8 image, (**b**) ground truth, (**c**) water body results.

**Table 5.** Accuracy evaluation after removing mountain shadows.

| Data | Accuracy (%) | Kappa |
|---|---|---|
| Water body before removing mountain shadows | 93.06 | 0.9173 |
| Water body after removing mountain shadows | 95 | 0.9315 |

### 4.3. Accuracy and Efficiency in the Flood Monitoring

The accuracy and kappa coefficients of the trained SVM model in the testing dataset were 97.77% and 0.9521, respectively, which proved its effectiveness in flood monitoring in

a large basin. However, the existence of mountains in the west of the MLB, which produce shadows similar to water bodies, hinders further accuracy improvement during flood monitoring. Therefore, we proposed a shadow removal model to remove the mountain shadows, and the accuracy and Kappa coefficient of flood monitoring after removing mountain shadows were improved by 1.94% and 0.0142, respectively. The rapid development of the floods and since floods are usually accompanied by cloudy or rainy weather, the accuracy assessment of flood monitoring results from optical remote sensing images is limited. Meanwhile, since the MLB is very large and there are many rivers and lakes, it is unrealistic to evaluate the accuracy of the SVM model and mountain shadow removal model in the basin. Therefore, we selected several testing regions to quantitatively evaluate the accuracy of the model, but this approach is uncertain. The uncertainty of flood monitoring hinders emergency monitoring of flood control. It is difficult to study the uncertainty of flood monitoring.

The purpose of flood monitoring research is the practical application of emergency monitoring. In the process of flood emergency monitoring in a large basin, the SVM model and mountain shadow removal model can be used in combination, which can quickly extract the spatial and temporal distribution of floods in a short time. This provides a scientific basis for flood early warning, disaster relief, and post-disaster assessment, and has high production efficiency.

*4.4. Inundation Analysis*

In order to evaluate the losses caused by this major watershed flood, the study superimposed land use data into the water body extraction results to analyze the specific categories of land use inundated by the floods. The land use data used in the study was produced in 2020, which was consistent with the time of the research subject, so as to avoid changes in land use status due to differences in data years. As shown in Figure 11 and Table 6, a variety of land uses were affected by the basin floods, and a total of 8526 km$^2$ of land was inundated, which was mainly distributed along the Yangtze River. Among them, cropland was the most severely affected, with an affected area of 6160 km$^2$, accounting for 72.25% of the total inundated area, with the inundated cropland mainly distributed along the main stream of the Yangtze River and around Poyang Lake. The floods severely damaged agricultural production and caused serious losses to the MLB. Wetland and forest were more seriously affected, and their inundated areas were 985 km$^2$ and 677 km$^2$, respectively. The inundated wetland was mainly distributed along the Dongting Lake and Poyang Lake, and the inundated forest was mainly distributed in the west and south of the study area. Grassland, buildings, and bare land were less affected, and their inundated areas were 416 km$^2$, 182 km$^2$, and 106 km$^2$, respectively, accounting for less than 5%. The inundated grassland was mainly located in the west and south of the study area, while the inundated buildings were mainly located along the Wuhan section of the Yangtze River and the Nanjing section of the Yangtze River.

**Table 6.** The statistics of inundated land by land use in the MLB.

| Inundated Type | Inundated Area (km$^2$) | Proportion (%) |
|---|---|---|
| Buildings | 182 | 2.14 |
| Forest | 677 | 7.94 |
| Grassland | 416 | 4.88 |
| Bare land | 106 | 1.24 |
| Wetland | 985 | 11.55 |
| Cropland | 6160 | 72.25 |

*4.5. Disaster Analysis of the Typical Lakes*

The MLB is vast, and there are many rivers and lakes [40], so it is difficult to analyze the disaster across the basin. Therefore, we selected three typical lakes for disaster analysis,

Taihu Lake, Poyang, Lake and East Dongting Lake, which are in the west, middle, and east of the study area, respectively.

**Figure 11.** The distribution map of inundated land by land use in the study area.

### 4.5.1. Taihu Lake

Taihu Lake is the third largest freshwater lake in China, which is in the south of Jiangsu Province, with the geographical range of 30°55′~31°32′ N and 119°52′~120°36′ E.

The measured water level data could supplement SAR images for flood monitoring [41]. Therefore, this paper collected daily water level data from the Dongting Xishan hydrological station located in Taihu Lake, as shown in Figure 12. It could be seen that the water level of Taihu Lake was 3.22 m on 22 June, when the water level rose rapidly at the beginning, crossing the warning water level of 3.8 m on 29 June and reaching the highest level of 4.75 m on 21 July. Then, the water level gradually decreased and dropped to the warning level on 13 August. Figures 13 and 14 showed the water area changes of Taihu Lake. It could be seen that the changes of the water body area were consistent with the trend of water level changes, and its water area increased from 2289 km$^2$ on 16 June to 2344 km$^2$ on 22 July, which was an increase of 55 km$^2$. Then, the water area decreased to 2307 km$^2$ on 27 August, which was a decreased of 37 km$^2$.

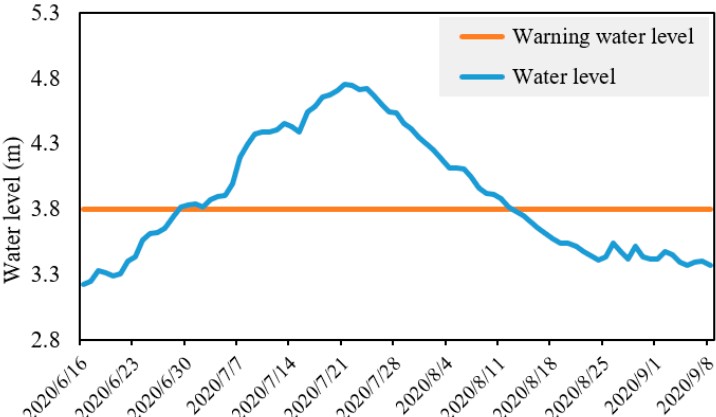

**Figure 12.** The water level process map of Taihu Lake. The water level data from the hydrological station were based on the Wusong elevation.

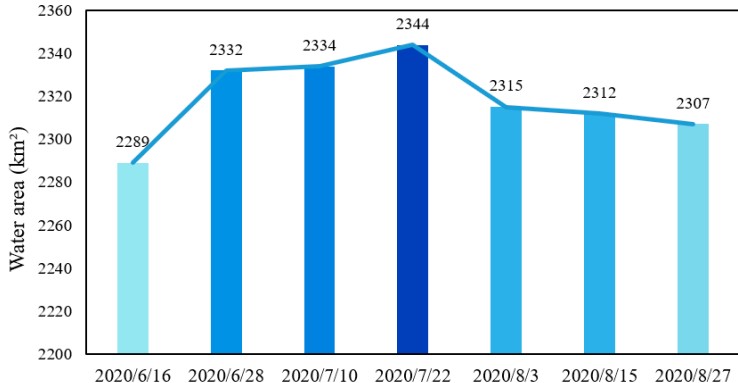

**Figure 13.** The water area changes of Taihu Lake.

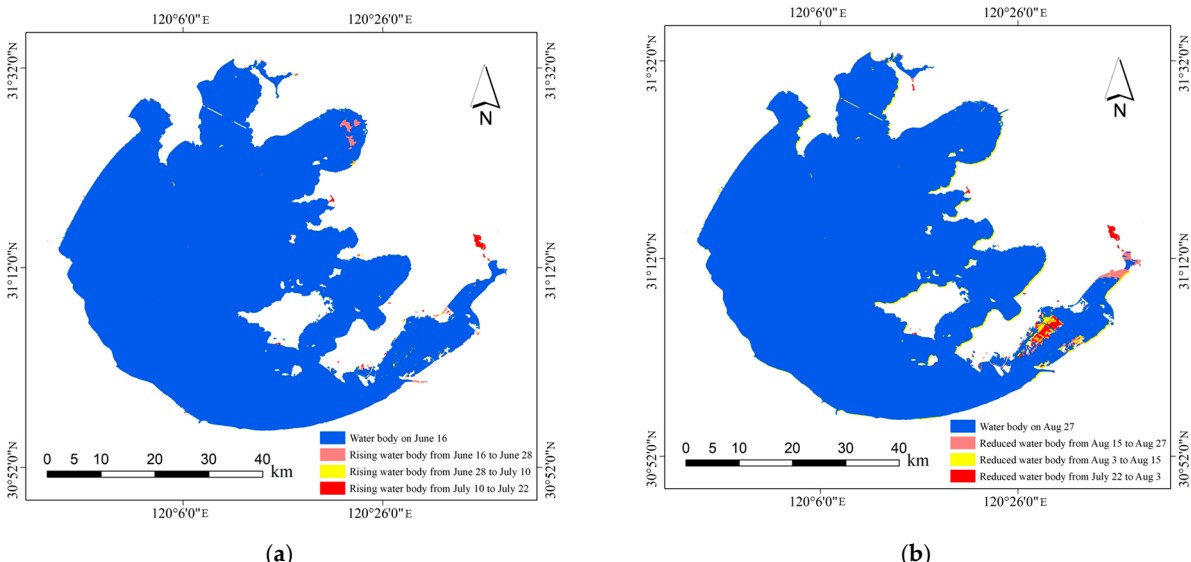

(**a**)　　　　　　　　　　　　　　　　　　　　　　　(**b**)

**Figure 14.** The spatial distribution maps of the water body in Taihu Lake. (**a**) The rising water stage, (**b**) the reduced water stage.

### 4.5.2. Poyang Lake

Poyang Lake is the largest freshwater lake in China, which is located in the north of Jiangxi Province, with the geographical range of 28°22′ N to 29°45′ N and 115°47′ E to 116°45′ E, and it plays an important role in flood storage and drought relief in the MLB [42]. The water area changes of Poyang Lake are shown in Figure 15. It could be seen that the water area of Poyang Lake showed a change trend of "steep rise and slow fall" during the whole flooding period. The water area of Poyang Lake first increased rapidly from 2832 km² on 16 June to 3794 km² on 20 July, and then decreased slowly to 3574 km² on 6 September due to the influence of high water levels of the Yangtze River. In terms of the temporal rate of change, from 16 June to 20 July, the water area increased by a total of 962 km² with a daily average increase of 40 km²/d. From 20 July to 6 September, the water area decreased by a total of 220 km² with a daily average decrease of 4.58 km²/d. Figure 16 shows the spatial distribution maps of the water body in the Poyang Lake. It could be seen that there was no flood in the west, southwest, and east of Poyang Lake on 26 June 2020. On 8 July 2020, there was obvious flooding in the southwest of Poyang Lake, and a large area of the Yellow Lake flood storage area was inundated. On 20 July 2020, there was obvious inundation in the west and east of Poyang Lake. The slow decrease of the water area brought great pressure to the flood control and disaster relief in the MLB.

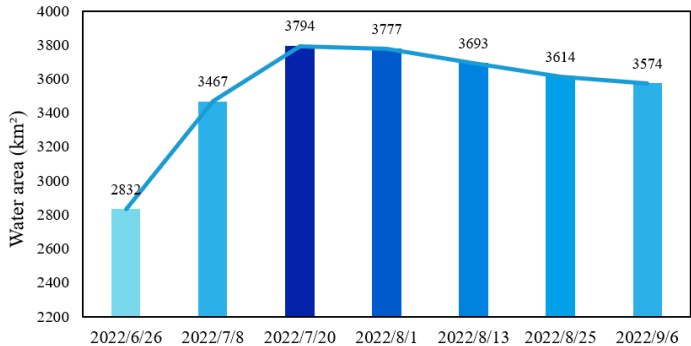

**Figure 15.** The water area changes of Poyang Lake.

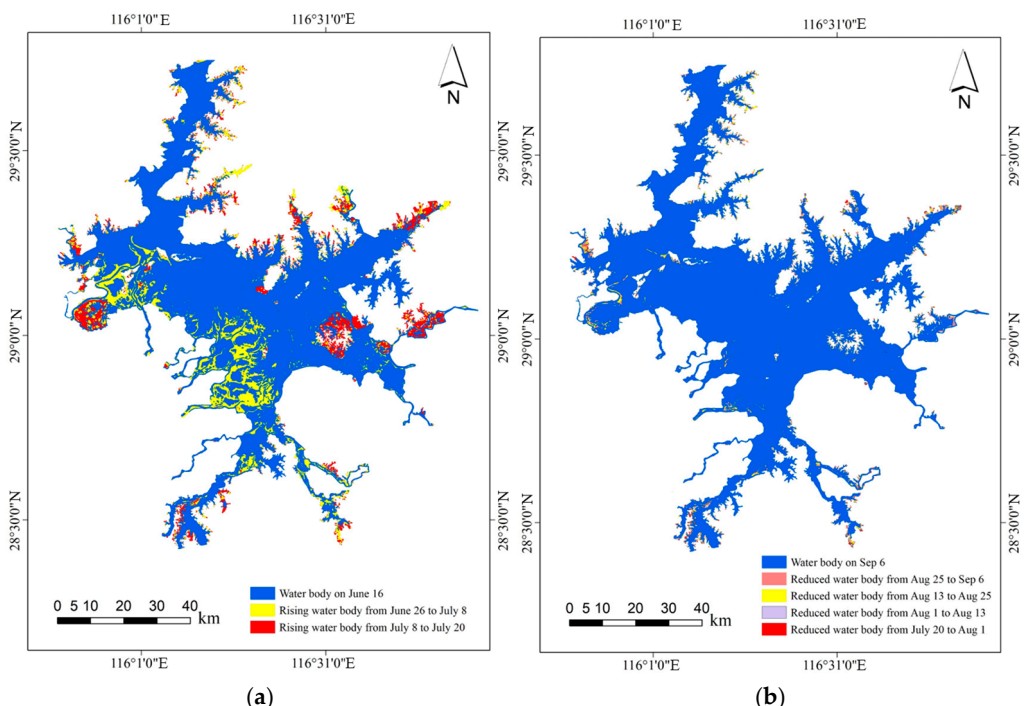

**Figure 16.** The spatial distribution maps of the water body in Poyang Lake. (**a**) The rising water stage, (**b**) the reduced water stage.

4.5.3. East Dongting Lake

Dongting Lake is the second largest freshwater lake in China, and it is located in the north of Hunan Province with the geographical range of 28°30′ N~30°20′ N and 112°25′ E~113°15′ E, and is an important storage lake in the Yangtze River basin. However, the SAR satellite did not observe the west of Dongting Lake at the early stage of flooding, and therefore the East Dongting Lake was selected for disaster analysis in this paper. Figure 17 shows the water area changes of East Dongting Lake, and the change trend was similar to that of Poyang Lake, that is, the change trend of "steep rise and slow fall". Its water area first increased rapidly from 1015 km² on 19 June to 1614 km² on 18 July, which was an water area increase of 599 km², and then the water area continued to increase slowly to 1629 km² on 30 July, after which the water area decreased slowly to 1480 km² on 4 September. Figure 18 shows the spatial distribution maps of the water body in East Dongting Lake from 19 June to 4 September 2020. It could be seen that at the beginning of the flood, the middle and south of the East Dongting Lake did not appear to be inundated. As the water level rose, small lakes located in the middle and south of the East Dongting Lake joined together and inundated a large exposed area. After that, the water area slowly declined but the inundated areas did not show any significant recession.

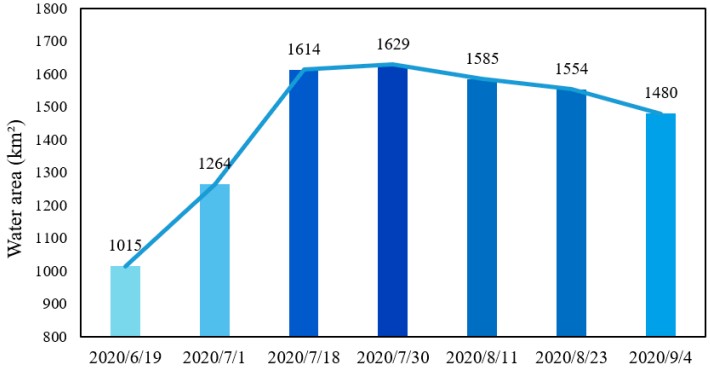

**Figure 17.** The water area changes of the East Dongting Lake.

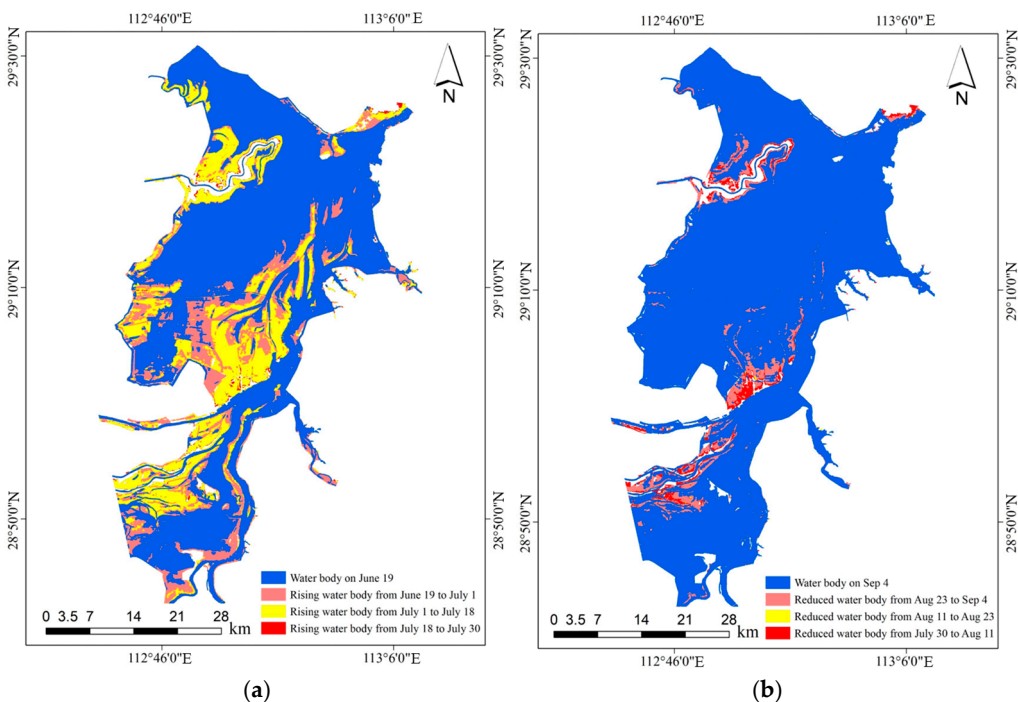

(**a**)            (**b**)

**Figure 18.** The spatial distribution maps of the water body in the East Dongting Lake. (**a**) The rising water stage, (**b**) the reduced water stage.

### 4.6. Limitations and Implications

Although this paper adopted the SVM model to extract the flood water body information on the GEE platform and proposed a function model to remove the influence of the mountain shadows, thereby further improving the accuracy of large-scale flood monitoring, there were still some shortcomings in this paper. The first was the single data source. This paper used Sentinel-1 images to monitor floods, which were difficult to use to meet the requirements for large-scale floods. In the future, multi-source remote sensing images such as GF-3 and Landsat 8 can be combined. The second was the linear model to remove the effect of mountain shadows. It is not ideal, and the model can be improved in the future to improve the removal rate of mountain shadows. In addition, with the development of machine learning algorithms, new algorithms are constantly being used for flood monitoring [14,43]. In the future, we can apply more algorithms to find the most suitable algorithm for flood monitoring. Finally, for large-scale watershed flooding, it is difficult to grasp the evolution trend and characteristics of floods. This paper only selected three typical lakes to analyze flood disasters. More lakes and rivers can be selected for research in the future.

## 5. Conclusions

Based on the GEE cloud platform and Sentinel-1 SAR images, this paper used the SVM model to extract flood water bodies during floods, and then analyzed the flood disaster situation in the MLB to solve the low accuracy of the extraction of large-scale watershed flood disasters and the difficulty of removing mountain shadows. The results showed that: (1) the evaluation index accuracy and kappa coefficient of the trained SVM model in the testing dataset were 97.77% and 0.9521, respectively. (2) Compared with the other three methods for removing mountain shadows, the linear function model proposed based on samples had the best effect, and its shadow recognition rate was 75.46%. Applying the function model to the flood water body extraction maps could mitigate the interference of mountain shadows. (3) We analyzed flood disasters based on multi-temporal flood water body extraction maps. The flood inundated a total of 8526 km$^2$ of land, of which cropland was the most severely damaged, accounting for 72.25% of the total inundated area. The flood seriously damaged the agricultural production in the MLB.

Remote sensing images are used as the record of surface information, and there are differences between the information extracted from images and actual surface information. Although the flood monitoring method in this paper has high accuracy, the bias of the SAR images for recording surface information, the error of flood information extraction, and the difficulty in carrying out accuracy assessment on a large scale, lead to the uncertainty of flood monitoring accuracy. We chose good quality remote sensing images and an intelligent algorithm to reduce this uncertainty. In the future, accuracy assessment can be carried out in more areas to reduce the accuracy uncertainty of flood monitoring.

The flood monitoring method and technical process in this paper can be used in actual flood monitoring, which has high production efficiency. It can provide important support for emergency response and disaster relief of relevant departments, and is of great significance to improve disaster emergency management capability, and provide important guarantees for subsequent studies such as flood development trend and post-disaster damage assessment.

**Author Contributions:** Conceptualization, Jingming Wang, Futao Wang, Shixin Wang, Yi Zhou, Qing Zhao and Longfei Liu; methodology, Jingming Wang and Futao Wang; software, Jingming Wang; validation, Jingming Wang, Jianwan Ji and Zhenqing Wang; formal analysis, Jingming Wang; investigation, Jingming Wang; resources, Futao Wang; data curation, Jingming Wang; writing—original draft preparation, Jingming Wang, Futao Wang, Jianwan Ji and Zhenqing Wang; writing—review and editing, Jianwan Ji; visualization, Jingming Wang; supervision, Futao Wang; project administration, Jingming Wang and Futao Wang; funding acquisition, Jingming Wang and Futao Wang. All authors have read and agreed to the published version of the manuscript.

**Funding:** This research was funded by the National Key R&D Program of China (2022YFC3006400, 2022YFC3006403), Fujian Provincial Science and Technology Plan Project (No. 2021T3065), and Finance Science and Technology Project of Hainan Province (No. ZDYF2021SHFZ103).

**Institutional Review Board Statement:** Not applicable.

**Informed Consent Statement:** Not applicable.

**Data Availability Statement:** The data presented in this study are available on request from the corresponding author.

**Acknowledgments:** We would like to thank the anonymous reviewers and the editors for their insightful comments and helpful suggestions.

**Conflicts of Interest:** The authors declare no conflict of interest.

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
