# Peer review of "Flood Monitoring in the Middle and Lower Basin of the Yangtze River Using Google Earth Engine and Machine Learning Methods"

_ijgi, doi:10.3390/ijgi12030129_

Round 1
Reviewer 1 Report
Dear Authors,
Please find my comments attached in the word document. Please take into account all the aspects pointed out and fully address them.
Best regards

Author Response
- L87 – Reconsider the phrase topic.
Response: We sincerely appreciated for professors’ constructive suggestions. We have modified this sentence (Line 87).
- L120-122 – you state that “We used 320 Sentinel-1 images with an imaging time from June 16, 2020 to September 8, 2020 to cover the entire flood stage.”, but you do not describe the stages of the floods. As previously mentioned in the general comments, this must be described in more detail.
Response: We sincerely appreciated for professors’ constructive suggestions. Five floods occurred in the Yangtze River basin in 2020, on July 2, July 12, July 26, Au-gust 14, and August 17. We chose images after June 16 that could cover the study area before flood, and images before September 8 to cover the study area after flood. This imaging time completely includes the three stages of the flood: pre-flood, peak-flood and post-flood. We have added the relevant description in the manuscript (Line 123 - Line 125).
- L131-133 – this phrase is confusing. You’ve talked about Sentinel-1 at L114-117, and now you repeat, but mix with Sentinel 2? This is ambiguous, must be rephrased.
Response: We sincerely appreciated for professors’ constructive suggestions. Line 135 in section 2.2.2 is duplicated with Line 115, and we have removed the duplicate description (Line 135).
- L293-294 – please describe more in a more detailed manner in the manuscript, how the erosion and dilation methods were applied to post-process the results.
Response: We sincerely appreciated for professors’ constructive suggestions. We have added the detailed description about morphological operations (Line 304 - Line 308).
- L335 – “The study selected” – this should be rephrased, because the study doesn’t do anything by “itself”.
Response: We sincerely appreciated for professors’ constructive suggestions. We have modified this sentence (Line 350).
- L376 – please define “ground objects”. Is it buildings? Is it number of entities, summed up? (Buildings, bridges, agricultural parcels etc?). If so, please detail. But, I would rather think you are not referring to “objects”, but rather to categories of land use, and you must clearly state this in the manuscript. And treat/mention buildings separately.
Response: We sincerely appreciated for professors’ constructive suggestions. “Ground objects” refers to land use, we have replaced “ground objects” with “land use” (Line 148, 388, 391, 392, 407, 408, 502).
- L383 – “the” – please use capital letter
Response: We sincerely appreciated for professors’ constructive suggestions. We have modified this sentence (Line 399).
- Figure 11 should be enlarged inside the manuscript, for a little better visual understanding of the content.
Response: We sincerely appreciated for professors’ constructive suggestions. We have enlarged this figure (Line 406).
- Figure 14, 16, 18 – the 7 stages are too similar, to be relevant in the manuscript. I would suggest making a multiple-data-frame map, with a single data-frame, which would be larger, in the middle, and then, several smaller data frames, with zoom-ins for the areas which actually reveal difference between the different image sets, so that the differences are actually visible.
Response: We sincerely appreciated for professors’ constructive suggestions. We have modified figure 14, 16 and 18 to see the differences (Line 433, 454, 474).
Reviewer 2 Report
Dear Editor
This paper addresses the applicability of the GEE platform and ML techniques for the extraction of flood bodies in the lower basin of the Yang-Tze. Overall, the paper is well-written and well-organized, But there are some minor tips that should be considered before publishing in this journal. These tips include:
1- It's recommended to use more relevant studies to this topic. This topic is a widely used approach, so the authors can find more similar studies.
2- It was better to use ground-based observations for assessing the accuracy of classification methods. For example, observed flood extent is the best one for this issue.
3- If it is possible, please estimate the flood damage values for each land cover types using depth-damage curves. I think it can be more interesting for readers if you do this issue.
4- English language and style need a double check prosses.
5- Please explain more about the Shadow removal model in the revised version.
Best
Author Response
- It's recommended to use more relevant studies to this topic. This topic is a widely used approach, so the authors can find more similar studies.
Response: We sincerely appreciated for professors’ constructive suggestions. We have read some studies and found some good algorithms for flood monitoring. We can compare more algorithms in the future to find the most suitable flood monitoring algorithm (Line 484 - Line 487).
- It was better to use ground-based observations for assessing the accuracy of classification methods. For example, observed flood extent is the best one for this issue.
Response: We sincerely appreciated for professors’ constructive suggestions. We do not have ground-based observations for assessing the accuracy. Ground-based observations are difficult to obtain. We usually use the real ground data obtained by optical remote sensing images interpretation to assess the accuracy of classification methods.
- If it is possible, please estimate the flood damage values for each land cover types using depth-damage curves. I think it can be more interesting for readers if you do this issue.
Response: We sincerely appreciated for professors’ constructive suggestions. Drawing the depth-damage curves needs in-situ water level data and empirical knowledge of land use loss, which are difficult to obtain. In the future, we can study the damage of flood inundated land use.
- English language and style need a double check prosses.
Response: We sincerely appreciated for professors’ constructive suggestions. We have revised the article in tense, the selection of words and sentence structures. We sincerely thank the professor for our suggestions.
- Please explain more about the Shadow removal model in the revised version.
Response: We sincerely appreciated for professors’ constructive suggestions. We have added the detailed description of the shadow removal model in the revised version (Line 261 – Line 263, Line 289 – Line 290, Line 295 – Line 298).
Reviewer 3 Report
It is an interesting paper that can certainly reveal areas affected by flooding, serving to quickly assess impacts.
Figure 1 could have a background so that the region does not appear as an island in the figure ; line 123 is missing an "L" in the word Sentinel; on line 225 FN must be FP (equation 2).
Author Response
- Figure 1 could have a background so that the region does not appear as an island in the figure.
Response: We sincerely appreciated for professors’ constructive suggestions. We have added a background to figure 1 (Line 111).
- Line 123 is missing an "L" in the word Sentinel.
Response: We sincerely appreciated for professors’ constructive suggestions. We have added the letter "L" to this word (Line 126).
- On line 225 FN must be FP (equation 2).
Response: We sincerely appreciated for professors’ constructive suggestions. We have checked equation 2 and modified it (Line 227).